# A Supervised Method for Nonlinear Hyperspectral Unmixing

**Bikram Koirala** [1,*], **Mahdi Khodadadzadeh** [2], **Cecilia Contreras** [2], **Zohreh Zahiri** [1], **Richard Gloaguen** [2] and **Paul Scheunders** [1]

[1] IMEC-Vision Lab, Department of Physics, University of Antwerp, 2000 Antwerp, Belgium; zohreh.zahiri@uantwerpen.be (Z.Z.); paul.scheunders@uantwerpen.be (P.S.)
[2] Helmholtz Institute Freiberg for Resource Technology, 09599 Freiberg, Germany; m.khodadadzadeh@hzdr.de (M.K.); i.contreras@hzdr.de (C.C.); r.gloaguen@hzdr.de (R.G.)
* Correspondence: bikram.koirala@uantwerpen.be

**Abstract:** Due to the complex interaction of light with the Earth's surface, reflectance spectra can be described as highly nonlinear mixtures of the reflectances of the material constituents occurring in a given resolution cell of hyperspectral data. Our aim is to estimate the fractional abundance maps of the materials from the nonlinear hyperspectral data. The main disadvantage of using nonlinear mixing models is that the model parameters are not properly interpretable in terms of fractional abundances. Moreover, not all spectra of a hyperspectral dataset necessarily follow the same particular mixing model. In this work, we present a supervised method for nonlinear spectral unmixing. The method learns a mapping from a true hyperspectral dataset to corresponding linear spectra, composed of the same fractional abundances. A simple linear unmixing then reveals the fractional abundances. To learn this mapping, ground truth information is required, in the form of actual spectra and corresponding fractional abundances, along with spectra of the pure materials, obtained from a spectral library or available in the dataset. Three methods are presented for learning nonlinear mapping, based on Gaussian processes, kernel ridge regression, and feedforward neural networks. Experimental results conducted on an artificial dataset, a data set obtained by ray tracing, and a drill core hyperspectral dataset shows that this novel methodology is very promising.

**Keywords:** hyperspectral unmixing; spectral mixing models; machine learning algorithms

## 1. Introduction

Spectral unmixing aims at estimating the fractional abundances of the different pure materials, so-called endmembers, that are contained within a hyperspectral pixel. The main assumption is that the captured reflectance spectrum of each hyperspectral pixel is a mixture of the reflectances of few endmembers. Based on that, for unmixing, a mixing model can be applied and the error between the true spectrum and the spectrum that is reconstructed by the applied mixing model is minimized [1,2].

Typically, spectral unmixing is performed by applying the linear mixing model (LMM). This model is valid only when every incoming ray of light interacts only once with a specific pure material in the pixel before reaching the sensor. Taking into account the physical non-negativity and sum-to-one constraints of the fractional abundances, the reconstruction error between the true spectrum and the linearly mixed spectrum can be minimized using the fully constrained least squares unmixing (FCLSU) procedure [3]. The LMM has shown very good performance in scenarios where the Earth's surface contains large flat areas with clearly separated regions containing different endmembers. However, the performance of LMM is not satisfactory for scenarios where the scene has a complex geometrical structure. In these scenarios, the incident ray of light may interact with several pure materials within

the pixel before reaching the sensor. This causes the captured reflectance spectra to be highly nonlinear mixtures of the endmember reflectances. To explain these nonlinearities, bilinear models have been proposed [4]. These models extend the LMM by adding bilinear terms, describing that the incident ray of light interacts with two pure materials within the pixel before reaching the sensor. Other nonlinear models try to extend these towards larger numbers of multiple interactions, e.g., the multilinear mixing model (MLM) [5] and the p-linear ($p > 2$) mixture model (pLMM) [6–8]. The most advanced nonlinear mixing models are physically-based radiative transfer models. These models are often employed for modeling intimate mixtures of materials. They represent the medium as a half-space filled with particles with known densities and distributions of physical attributes. The Hapke model is a simplified version of a radiative transfer model, [4,9,10].

The available nonlinear mixing models either oversimplify the complex interactions between light and materials or, by considering different kinds of interactions, tend to be inherently complex and non-invertible. Also, all pixels do not necessarily follow the same particular model. Moreover, the parameters of nonlinear mixing models are generally hard to interpret and link to the actual fractional abundances.

Instead of depending on a specific nonlinear mixing model, a new set of algorithms was developed that performed the unmixing of hyperspectral imagery in a reproducing kernel Hilbert space. In [11], FCLSU was kernelized by radial basis function kernels (KFCLS). However, no improvement in the unmixing result of intimate mixtures was observed [12].

Another problem is that when using a mixing model, it is assumed that the endmembers are known, i.e., obtained as pure pixels from the data. However, in real scenarios, the pure pixel assumption may collapse due to the relatively low spatial resolution (e.g., ground sampling distances of 30–60m) of hyperspectral images. Many methods have been proposed to solve this problem. Methods that estimate the endmembers along with the fractional abundances have been proposed, that in particular handle the situations when no pure pixels are present in the data, such as volume-constrained nonnegative matrix factorization (MVCNMF) [13], minimum volume simplex analysis [14] and robust collaborative nonnegative matrix factorization [15]. All these methods assume that endmembers span the corner of a linear simplex, and thus cannot cope with nonlinearities. To tackle nonlinearities, in [16], nonlinear kernelized NMF was presented. More recently, unmixing based on autoencoders [17–22] are employed to estimate endmembers and fractional abundances simultaneously. However, these algorithms are either limited to the linear mixing model or implementation of an existing nonlinear (bilinear) model to the autoencoder framework. Finally, sparse unmixing techniques [23–25] estimate the fractional abundances of the mixed pixel by linear combinations of spectra from spectral libraries e.g., collected on the ground by a field spectroradiometer. Again, these methods inherently apply the linear mixing model, and cannot cope with nonlinearities.

Finally, the use of fixed endmember spectra results in significant estimation errors [26] due to spectral variability caused by spatial and temporal variability and variations in illumination conditions. One of the first attempts to address this issue was made by introducing the multiple endmember spectral mixture analysis (MESMA) technique [27]. Since then, several spectral unmixing techniques that consider endmember variability have been developed and can be classified into three categories: algorithms based on endmember bundles, computational models, and parametric physics-based models [28]. All these methods, however, inherently apply the linear mixture model.

Alternatively to the assumption of a particular mixing model, some attempts have been made to learn the complex mixing effects using a completely data-driven approach [29–31]. We will refer to such methods as supervised spectral unmixing methods, since they require ground truth training data, in the form of the actual compositions for a number of pixels (i.e., the endmember spectra and the fractional abundances). The main strategy of these techniques is to learn a map from the spectra to the fractional abundances. For example, in [31], a multi-layer perceptron (MLP) was applied as a multivariate regression technique to learn a map from reflectance spectra to the fractional abundances. Similarly, in [32], support vector regression (SVR) was used. In [33], an auto-associative neural

network was used to extract low dimensional features from the hyperspectral image. The reduced measurement vector was further used as an input to an MLP scheme for pixel-based fuzzy classification. To tackle the endmember variability, in [34], the mapping was performed using a multi-task Gaussian process framework.

One major drawback of this mapping strategy is that the estimated fractional abundances do not comply with the physical positivity and sum-to-one constraints. This causes substantial amounts of abundance values to become lower than 0 or larger than 1, leading to a loss of the physical meaning of the estimated parameters [32]. Moreover, most of the methods either use pure pixels or linearly mixed pixels as training data, leading to performances that cannot outperform the simple LMM [35]. In order to really validate the potential of these methods, actual ground truth abundances of nonlinearly mixed spectra should be provided.

In this work, we propose a strategy to solve these issues. In the proposed method, first linearly mixed spectra are generated from the endmembers and their fractional abundances from the available training data. Then, a map between the actual training spectra and the generated linear spectra is learned by using a regression technique. Finally, after mapping the unknown spectra using the learned regression model, the FCLSU technique is applied to estimate the fractional abundances of the mapped spectra. In order to capture the nonlinearities in the learned maps, the availability of a training set of nonlinearly mixed spectral reflectances and ground-truth information about their endmembers and fractional abundances is a prerequisite.

The proposed unmixing strategy has a number of advantages compared to the state of the art. First of all, it accounts for the nonlinear behavior of spectral reflectances, which is expected to be high in scenes that contain complex geometrical structures or mixtures of minerals. Second, the method avoids dependency on a specific model. Third, the generalization of the mapping procedure allows to learn nonlinearities of different nature simultaneously and makes it robust to noise. Finally, opposite to a direct unconstrained mapping of the spectra to abundances by regression, the approach allows for a straightforward estimation of the fractional abundances with a clear physical meaning. In order to do so, our approach does require the endmember spectra. However, the generalization of the mapping can handle deviations between the applied endmembers and the true ones. As a result, in case that endmember spectra are not available, e.g., when there are no pure pixels present in the data, the endmembers can be collected from a spectral library, e.g., the USGS spectral library.

The mapping to the linear model can be applied by any nonlinear regression technique. In our previous work [36], this idea was first applied by using an MLP. In this work, we further elaborate on this idea by using different multivariate regression methods for mapping and reporting extensive experimental results. More specifically three mapping methods are presented, based on neural networks (NN), kernel ridge regression (KRR) [37,38], and Gaussian processes (GP) [39]. The presented methodology is validated on an artificial mineral dataset, a synthetic orchard scene, and a real drill core hyperspectral dataset, for which high-quality nonlinear ground reference data is available. The method is compared to some nonlinear mixing models and unconstrained direct mapping onto the abundances.

The remaining of the paper is organized as follows: In Section 2, we describe the proposed strategy along with the three different mapping approaches. In Section 3, we describe the datasets on which we validated the proposed method. In Section 4, we present the experimental results, followed by a discussion in Section 5. Section 6 concludes this work.

## 2. Methodology

Figure 1 illustrates the proposed methodology. In the first step, the endmembers and corresponding ground truth fractional abundances of a training set are combined to generate linearly mixed spectra. Then, a multivariate regression technique is applied to map nonlinear spectra to these linearly mixed spectra. Finally, linear unmixing is applied to estimate the fractional abundances of the mapped spectra. In the following, we explain these steps in more details.

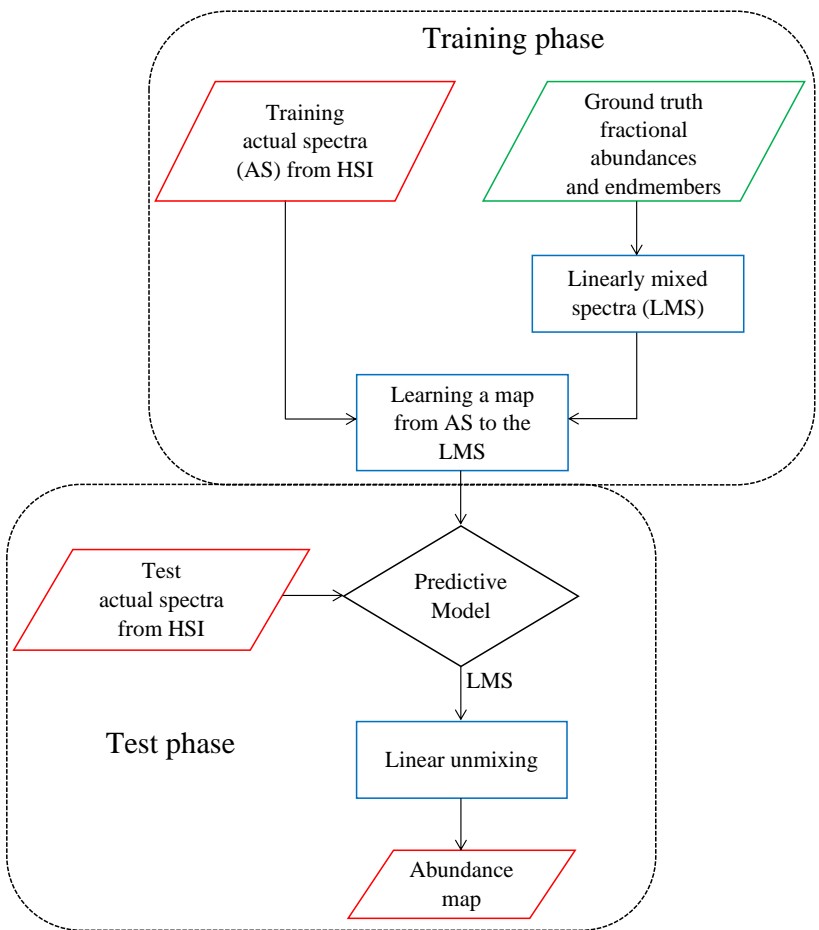

**Figure 1.** Flowchart of the proposed method. HSI refers to hyperspectral image.

### 2.1. Generating Linearly Mixed Training Spectra

Let $\mathbf{E}(\{\mathbf{e}_j\}_{j=1}^{p} \in \mathbf{R}_{+}^{d})$ be a set of $p$ endmembers (i.e., pure spectra) composed of $d$ spectral bands, and $\mathbf{A}(\{\mathbf{a}_i\}_{i=1}^{N} \in \mathbf{R}_{+}^{p})$ denote a matrix of fractional abundances of the endmembers $\mathbf{E}$ from $N$ samples. With the aforementioned definitions in mind, we assume that the hyperspectral pixels $\mathbf{Y}(\{\mathbf{y}_i\}_{i=1}^{N} \in \mathbf{R}_{+}^{d})$ are generated by a nonlinear function $F$ of the endmembers and fractional abundances:

$$\mathbf{y}_i = F(\mathbf{E}, \mathbf{a}_i) + \boldsymbol{\eta}_i, \tag{1}$$

where $\boldsymbol{\eta}_i$ represents Gaussian noise. Each nonlinear mixing model corresponds to a particular choice of $F$. Instead of depending on a particular mixing model, we propose a supervised method to learn $F$. To this aim, we assume that a set of $n$ training samples with known fractional abundances: $\mathcal{D} = \{(\mathbf{y}_1, \mathbf{a}_1), \ldots, (\mathbf{y}_n, \mathbf{a}_n)\}$ is available. From these abundances and the endmembers $\mathbf{E}$, linearly mixed training spectra $\mathbf{X}_{\mathcal{D}} = \{\mathbf{x}_i\}_{i=1}^{n}$ are generated:

$$\mathbf{x}_i = \mathbf{E}\mathbf{a}_i, \forall i \in \{1, \ldots, n\}. \tag{2}$$

### 2.2. Mapping

The second step of the proposed method is to learn a map from the nonlinear training spectra $\mathbf{Y}_{\mathcal{D}} = \{\mathbf{y}_i\}_{i=1}^{n}$ to the generated linearly mixed spectra $\mathbf{X}_{\mathcal{D}}$. The learning of this map can be performed in different ways. In this work, we choose one method based on NN and two state-of-the-art machine learning regression algorithms, i.e., KRR, and GP. After learning the map, the obtained regression model is used to map the test nonlinear spectra $\mathbf{Y}_t = \{\mathbf{y}_t\}_{t=n+1}^{N}$ to linearly mixed spectra $\mathbf{X}_t = \{\mathbf{x}_t\}_{t=n+1}^{N}$.

### 2.2.1. Mapping Using NN

The first method uses a a feedforward NN, a multilayer perceptron (MLP) to perform the mapping. In this network, the input layer contains $d$ nodes, representing the spectral bands of the actual spectra $\mathbf{Y}_\mathcal{D}$. There is one hidden layer consisting of $h = 10$ nodes (as in [36]) and the output layer of the same size as the input layer, containing the linearly mixed spectra $\mathbf{X}_\mathcal{D}$. The hyperbolic tangent function $\left(\tanh(a) = \frac{\exp(a) - \exp(-a)}{\exp(a) + \exp(-a)}\right)$ is used as an activation function for the hidden layer and the identity activation ($f(a) = a$) for the output layer. The architecture of the network is shown in Figure 2.

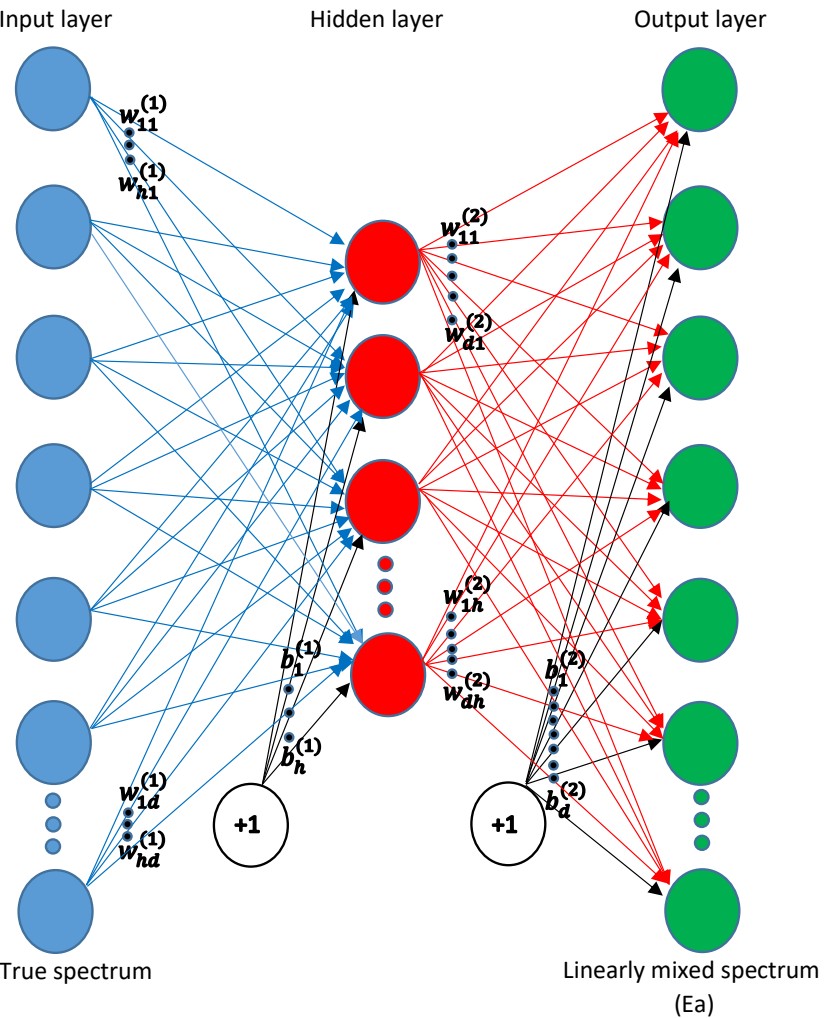

**Figure 2.** The feedforward neural network that is applied for the mapping.

The estimated map of the nonlinear spectra $\mathbf{Y}_t$ to the linear spectra $\mathbf{X}_t$ is then given by:

$$
\begin{aligned}
\mathbf{X}_t &= \begin{bmatrix} w^{(2)}_{11} & \cdots & w^{(2)}_{1h} \\ \vdots & \ddots & \vdots \\ w^{(2)}_{d1} & \cdots & w^{(2)}_{dh} \end{bmatrix} \tanh\left(\begin{bmatrix} w^{(1)}_{11} & \cdots & w^{(1)}_{1d} \\ \vdots & \ddots & \vdots \\ w^{(1)}_{h1} & \cdots & w^{(1)}_{hd} \end{bmatrix} \mathbf{Y}_t + \begin{bmatrix} b^{(1)}_1 \\ \vdots \\ b^{(1)}_h \end{bmatrix}\right) + \begin{bmatrix} b^{(2)}_1 \\ \vdots \\ b^{(2)}_d \end{bmatrix} \\
&= \mathbf{W}^{(2)} \tanh\left(\mathbf{W}^{(1)} \mathbf{Y}_t + \mathbf{b}^{(1)}\right) + \mathbf{b}^{(2)}
\end{aligned}
\tag{3}
$$

where the matrix $\mathbf{W}^{(1)}$ and the vector $\mathbf{b}^{(1)}$ are the weights and biases from input to hidden layer and the matrix $\mathbf{W}^{(2)}$ and the vector $\mathbf{b}^{(2)}$ are the weights and biases from hidden to output layer.

To train the network, we split the training dataset further into a training subset $n_1$ and a validation subset $n_2$ ($n_1 + n_2 = n$). In this work, we choose $n_1 = 0.9n$. The Levenberg–Marquardt

backpropagation algorithm [40] was used as the training algorithm. The network parameters (weights and biases) were estimated by using the training subset and minimizing the mean squared error loss function ($\frac{1}{n_1} \sum_{i=1}^{n_1} \left\| \mathbf{x}_i - \mathbf{W}^{(2)} \tanh \left( \mathbf{W}^{(1)} \mathbf{y}_i + \mathbf{b}^{(1)} \right) - \mathbf{b}^{(2)} \right\|^2$). To avoid overfitting, the validation subset is used to minimize the generalization error.

### 2.2.2. Mapping Using KRR

Ridge regression finds a linear relationship between the input $\mathbf{Y}_{\mathcal{D}}$ and output $\mathbf{X}_{\mathcal{D}}$ (i.e., $\{\mathbf{x}_i = \mathbf{w}^T \mathbf{y}_i\}_{i=1}^n$). For this purpose, the quadratic cost function that is regularized by the norm of $\mathbf{w}$ is minimized:

$$C = 1/2 \left( \lambda \|\mathbf{w}\|^2 + \left\| \mathbf{X}_{\mathcal{D}} - \mathbf{w}^T \mathbf{Y}_{\mathcal{D}} \right\|^2 \right), \tag{4}$$

where $\lambda$ is the regularization parameter. Minimizing 4 leads to:

$$\mathbf{w} = \left( \mathbf{Y}_{\mathcal{D}} \mathbf{Y}_{\mathcal{D}}^T + \lambda \mathbf{I} \right)^{-1} \left( \mathbf{Y}_{\mathcal{D}} \mathbf{X}_{\mathcal{D}}^T \right), \tag{5}$$

where $\mathbf{I}$ is the identity matrix. The above equation can be re-arranged to:

$$\mathbf{w} = \mathbf{Y}_{\mathcal{D}} \left( \mathbf{Y}_{\mathcal{D}}^T \mathbf{Y}_{\mathcal{D}} + \lambda \mathbf{I} \right)^{-1} \mathbf{X}_{\mathcal{D}}^T, \tag{6}$$

and the mapping of a nonlinear spectrum $\mathbf{y}_t$ to the linear spectrum $\mathbf{x}_t$ is obtained by:

$$\mathbf{x}_t = \mathbf{X}_{\mathcal{D}} \left( \mathbf{Y}_{\mathcal{D}}^T \mathbf{Y}_{\mathcal{D}} + \lambda \mathbf{I} \right)^{-1} \mathbf{Y}_{\mathcal{D}}^T \mathbf{y}_t \tag{7}$$

To allow for nonlinear mappings, ridge regression is kernelized [37,38] as:

$$\mathbf{x}_t = \mathbf{X}_{\mathcal{D}} \left( K(\mathbf{Y}_{\mathcal{D}}, \mathbf{Y}_{\mathcal{D}}) + \lambda \mathbf{I} \right)^{-1} K(\mathbf{Y}_{\mathcal{D}}, \mathbf{y}_t) \tag{8}$$

where $K(\mathbf{Y}_{\mathcal{D}}, \mathbf{y}_t)$ is the vector of kernel functions between the $n$ training points and a test sample and $K(\mathbf{Y}_{\mathcal{D}}, \mathbf{Y}_{\mathcal{D}})$ is the matrix of kernel functions between the $n$ training points. The kernel function used in this study is the radial basis function (RBF) kernel:

$$k(\mathbf{y}_i, \mathbf{y}_t) = \exp \left( - \frac{\|\mathbf{y}_i - \mathbf{y}_t\|^2}{2\sigma^2} \right). \tag{9}$$

Equation (8) involves computing the inversion of the ($n \times n$) kernel matrix $K$ that is regularized by $\lambda$. During the training phase, only the regularization parameter $\lambda$ and the parameter of the kernel ($\sigma$) need to be tuned. The tuning of hyperparameters of the KRR was done by 10-fold cross-validation with a grid search [41]. To determine the optimal pair ($\hat{\sigma}, \hat{\lambda}$), all possible combinations of $\sigma \in \{2^{-15}, \cdots, 2^3\}$ and $\lambda \in \{2^{-15}, \cdots, 2^5\}$ were applied and the average mapping error was calculated. For each test sample, the only computation involved is the determination of the kernel function between the test sample and the $n$ training samples.

### 2.2.3. Mapping Using GP

A GP [39] learns the nonlinear relationship between the input $\mathbf{Y}_{\mathcal{D}}$ and output $\mathbf{X}_{\mathcal{D}}$ as a Bayesian regression, by estimating the distribution of mapping functions that are coherent with the training set. It is assumed that the observed output variables $(\mathbf{x}_i)$ can be described as a function of the input $(\mathbf{y}_i)$:

$$\mathbf{x}_i = f(\mathbf{y}_i) = \boldsymbol{\phi}(\mathbf{y}_i)^T \mathbf{w}, \tag{10}$$

with prior $\mathbf{w} \sim \mathcal{N}(\mathbf{0}, \Sigma_d)$. The function $\phi(\cdot)$ maps the input to an infinite dimensional feature space. The mean and covariance of the outputs are given by:

$$\begin{aligned}
\mathbb{E}[f(\mathbf{y}_i)] &= \boldsymbol{\phi}(\mathbf{y}_i)^T \mathbb{E}[\mathbf{w}] = \mathbf{0} \\
\mathbb{E}[f(\mathbf{y}_i)f(\mathbf{y}_j)] &= \boldsymbol{\phi}(\mathbf{y}_i)^T \mathbb{E}[\mathbf{w}\mathbf{w}^T]\boldsymbol{\phi}(\mathbf{y}_j) = \boldsymbol{\phi}(\mathbf{y}_i)^T \Sigma_d \boldsymbol{\phi}(\mathbf{y}_j).
\end{aligned} \tag{11}$$

GP assumes that the covariance of the outputs can be modeled by the squared exponential kernel function:

$$\boldsymbol{\phi}(\mathbf{y}_i)^T \Sigma_d \boldsymbol{\phi}(\mathbf{y}_j) = k(\mathbf{y}_i, \mathbf{y}_j) = \sigma_f^2 \exp\left(-\sum_{b=1}^{d} \frac{(y_i^b - y_j^b)^2}{2l_b^2}\right), \tag{12}$$

where $\sigma_f^2$ is the variance of the input spectra, and $l_b$ is a characteristic length-scale for each band.

The joint distribution of the training output $(\mathbf{X}_{\mathcal{D}})$ and the test output $(f(\mathbf{Y}_t))$ can then be written as follows:

$$\begin{aligned}
p(f(\mathbf{Y}_t^T), \mathbf{X}_{\mathcal{D}}^T) &\sim \mathcal{N}\left(\mathbf{0}, \begin{bmatrix} K(\mathbf{Y}_t, \mathbf{Y}_t) & K(\mathbf{Y}_t, \mathbf{Y}_{\mathcal{D}}) \\ K(\mathbf{Y}_{\mathcal{D}}, \mathbf{Y}_t) & K(\mathbf{Y}_{\mathcal{D}}, \mathbf{Y}_{\mathcal{D}}) + \sigma_n^2 \mathbf{I} \end{bmatrix}\right) \\
&= \mathcal{N}\left(\mathbf{0}, \begin{bmatrix} \Sigma_{11} & \Sigma_{12} \\ \Sigma_{21} & \Sigma_{22} \end{bmatrix}\right),
\end{aligned} \tag{13}$$

where $\sigma_n^2$ is the noise variance of the training spectra, $K(\mathbf{Y}_{\mathcal{D}}, \mathbf{Y}_t)$ is the matrix of kernel functions between the $n$ training samples and the test samples, and $K(\mathbf{Y}_t, \mathbf{Y}_t)$ is the matrix of kernel functions between the test samples.

When using the partitioned inverse formula:

$$\begin{aligned}
&\begin{pmatrix} \Sigma_{11} & \Sigma_{12} \\ \Sigma_{21} & \Sigma_{22} \end{pmatrix}^{-1} \\
&= \begin{pmatrix} \Sigma^{-1} & -\Sigma^{-1}\Sigma_{12}\Sigma_{22}^{-1} \\ -\Sigma_{22}^{-1}\Sigma_{21}\Sigma^{-1} & \Sigma_{22}^{-1} + \Sigma_{22}^{-1}\Sigma_{21}\Sigma^{-1}\Sigma_{12}\Sigma_{22}^{-1} \end{pmatrix},
\end{aligned} \tag{14}$$

with $\Sigma = \Sigma_{11} - \Sigma_{12}\Sigma_{22}^{-1}\Sigma_{21}$, (13) can be factorized into the predictive distribution $p(f(\mathbf{Y}_t^T)\mathbf{X}_{\mathcal{D}}^T)$ and the marginal $p(\mathbf{X}_{\mathcal{D}}^T)$:

$$\begin{aligned}
p(f(\mathbf{Y}_t^T), \mathbf{X}_{\mathcal{D}}^T) &= p(f(\mathbf{Y}_t^T)\mathbf{X}_{\mathcal{D}}^T) p(\mathbf{X}_{\mathcal{D}}^T) \\
&= \mathcal{N}(\Sigma_{12}\Sigma_{22}^{-1}\mathbf{X}_{\mathcal{D}}^T, \Sigma)\mathcal{N}(\mathbf{0}, \Sigma_{22}).
\end{aligned} \tag{15}$$

The estimated map of the nonlinear spectra $\mathbf{Y}_t$ to the linear spectra $\mathbf{X}_t$ is then given by:

$$\begin{aligned}
\mathbf{X}_t &= f(\mathbf{Y}_t) = \mathbf{X}_{\mathcal{D}}\Sigma_{22}^{-1}\Sigma_{12}^T \\
&= \mathbf{X}_{\mathcal{D}}(K(\mathbf{Y}_{\mathcal{D}}, \mathbf{Y}_{\mathcal{D}}) + \sigma_n^2 \mathbf{I})^{-1} K(\mathbf{Y}_t, \mathbf{Y}_{\mathcal{D}})^T.
\end{aligned} \tag{16}$$

The hyperparameters of the kernel function in (12) are optimized by minimizing the log marginal likelihood of the training dataset $\log(p(\mathbf{X}_{\mathcal{D}}^T \mathbf{Y}_{\mathcal{D}}^T))$.

*2.3. Linear Unmixing*

Once the mapping is learned and the test spectra are mapped onto the linear spectra, the final step is to obtain the fractional abundances from the mapped linear spectra, by inverting the LMM model. The LMM model assumes that a pixel spectrum $(\mathbf{x}_i)$ can be reconstructed as a linear combination of the endmember spectra:

$$\mathbf{x}_i = \sum_{j=1}^{p} a_j \mathbf{e}_j + \boldsymbol{\eta}_i = \mathbf{E}\mathbf{a}_i + \boldsymbol{\eta}_i, \tag{17}$$

where $a_j$ is the fractional abundance of endmember $\mathbf{e}_j$. Because the fractional abundance is the volume percentage, it is assumed that no endmember can have a negative volume, yielding the abundance nonnegativity constraint (ANC) and that the observed spectrum are completely decomposed by endmember contributions, leading to the abundance sum-to-one constraint (ASC). The fully constrained least squares unmixing algorithm (FCLSU) considers both ANC and ASC [3], and estimates the fractional abundances by minimizing $\|\mathbf{x}_i - \mathbf{E}\mathbf{a}_i\|^2$ s.t. $\sum_j a_j = 1, \forall j : a_j \geq 0$. From a geometric point of view, endmembers span the corners of a linear simplex $S_p$ in $(p\text{-}1)$ dimensions and all spectra lie within the linear simplex.

## 3. Eperimental Data

The following datasets were applied in the validation:

*3.1. Dataset 1: Simulated Mineral Dataset*

The first dataset is generated by taking mineral endmembers from the USGS spectral library. These spectra contain 224 reflection values for wavelengths in the range of 383–2508 nm. Ground truth fractional abundances were generated uniformly and randomly from the unit simplex. Then, (non)linear spectra were artificially generated by applying specific mixture models. For this dataset, obviously ground truth is available.

In the next section, this simulated data will be used for different experiments, e.g., validating the method, comparing it against the direct mapping strategy, for its robustness to noise and to the number of endmembers. For each experiment, different data, using different models with varying noise levels, endmembers etc. will be generated.

*3.2. Dataset 2: Ray Tracing Vegetation Dataset*

This dataset is a synthetic orchard scene [42] containing mixtures of three endmembers: soil, weed patches, and citrus trees. To generate this orchard scene, a fully calibrated virtual citrus orchard ray tracer, developed in [43] was used. This scene contains $20 \times 20$ pixels with a spatial resolution of 2 m and 216 spectral bands in the range 350–2500 nm. After removing 31 water absorption bands (1330:1440 nm, 1770:1950 nm), 185 bands remained. For this dataset, the exact per pixel fractional abundances and the spectra of the endmembers are available. The endmember spectra of soil, weed, and citrus tree are shown in Figure 3.

*3.3. Dataset 3: Drill Core Hyperspectral Dataset*

A hyperspectral image [44] was acquired from a drill core sample by using a SisuRock drill core scanner, equipped with an AisaFenix VNIR-SWIR hyperspectral sensor. An RGB image of this drill core sample is shown in Figure 4a. The spectral range of the hyperspectral camera extends from 380 to 2500 nm, with a spectral resolution of 3.5 nm and 12 nm in the VNIR and SWIR respectively, resulting in 450 bands. The first 50 noisy bands are removed, leading to 400 bands between 537.76–2485.90 nm. The spatial resolution of the hyperspectral image is 1.5 mm/pixel.

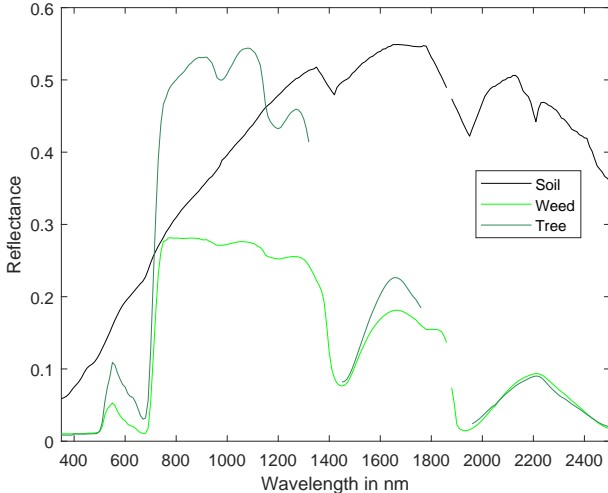

**Figure 3.** Endmember spectra of soil, weed, and citrus tree of the ray tracing experiment.

A thin section (about 39 × 29 mm and 30 μm thick) of the drill core sample (see red rectangle in Figure 4a) was polished. From this, mineralogical data of high spatial resolution (with a ground sampling distance of a few μm) were acquired by using scanning electron microscope (SEM)—mineral liberation analysis (MLA) technique. In this technique, the MLA software identifies minerals based on the combination of high-resolution back-scattered electron (BSE) image analysis and X-ray count rate information. An MLA image of high spatial resolution (3 μm/pixel) was obtained (see Figure 4b). The MLA image was resampled and co-registered with the hyperspectral image, after which ground truth fractional abundance maps of 373 pixels were generated. For more information about SEM-MLA and the generation of the groundtruth, we refer to [44].

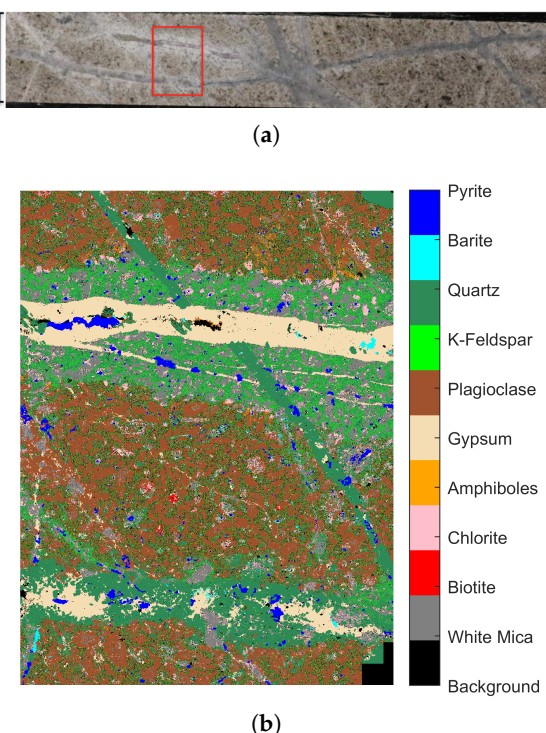

**Figure 4.** (**a**) RGB image of the drill core sample. The red rectangle represents the area where the ground truth fractional abundance maps were obtained by the scanning electron microscope (SEM)—mineral liberation analysis (MLA) analysis; (**b**) MLA image.

Due to the complexity of the sample and the resolution of the dataset, there are no pure pixels composed of only one mineral. For this reason, the endmembers of White Mica, Biotite, Chlorite, Amphiboles, Gypsum, Plagioclase, K-Feldspar, Quartz, Barite, and Pyrite were selected from the USGS spectral library, and interpolated to 400 bands between 537.76–2485.90 nm to make them compatible with the wavelength range of the AisaFenix VNIR-SWIR hyperspectral sensor (see Figure 5).

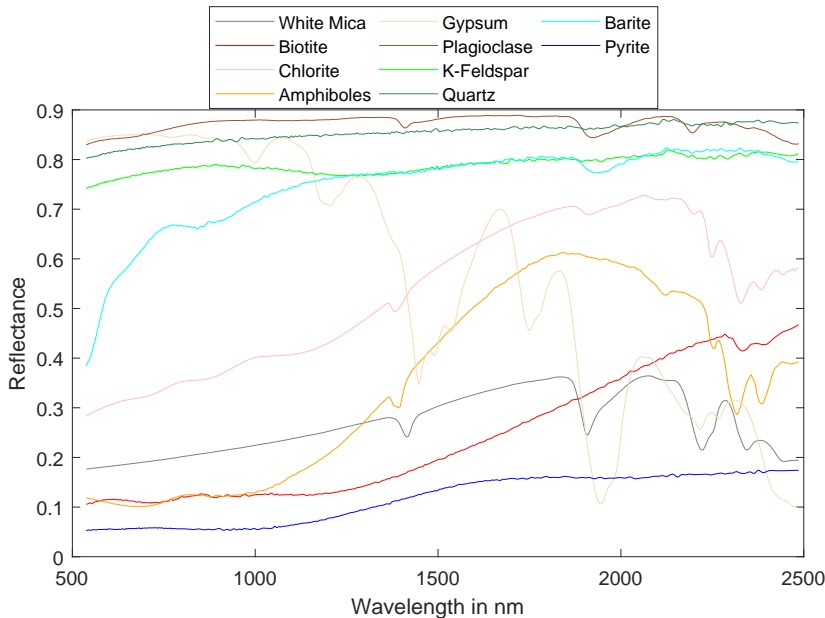

**Figure 5.** Spectra of ten mineral endmembers from the USGS spectral library.

## 4. Experiments

### 4.1. Experimental Set-Up

In the next section, the proposed unmixing method will be validated. To learn the mapping onto the linear model, we will apply the three different mapping methods and refer to the three unmixing methods as GP_LM, KRR_LM, and NN_LM.

#### 4.1.1. Comparison to Unsupervised Unmixing Models

We compared to the following 5 unsupervised spectral unmixing models:

- LMM, the linear mixing model
- Fan, the Fan model [4], a bilinear model
- PPNM, the polynomial post-nonlinear model [4], a bilinear model
- MLM, a multilinear mixing model [5]
- Hapke, the Hapke model, an intimate mixture model

See Table 1 for parameters and constraints of these models.

**Table 1.** Linear and nonlinear mixing models and their parameters. The assumptions for all models: $\forall m : a_m \geq 0, \sum_m a_m = 1$.

| Model | Equation | Parameters |
|---|---|---|
| Linear | $\mathbf{x} = \sum_{i=1}^{p} a_i \mathbf{e}_i$ | |
| FM | $\mathbf{x} = \mathbf{y} + \sum_{m=1}^{p-1} \sum_{k=m+1}^{p} b_{mk} \mathbf{e}_m \odot \mathbf{e}_k$ <br> $\mathbf{y} = \sum_{i=1}^{p} a_i \mathbf{e}_i$ | $\forall m \geq k : b_{mk} = 0$ <br> $\forall m < k : b_{mk} = a_m a_k$ |
| PPNM | $\mathbf{x} = \mathbf{y} + b(\mathbf{y} \odot \mathbf{y})$ <br> $\mathbf{y} = \sum_{i=1}^{p} a_i \mathbf{e}_i$ | $\forall m, k : b_{mk} = b a_m a_k$ <br> $b \in [-0.25, 0.25]$ |
| MLM | $\mathbf{x} = \dfrac{(1-P)\mathbf{y}}{1 - P\mathbf{y}}, \mathbf{y} = \sum_{i=1}^{p} a_i \mathbf{e}_i$ | $P \in [0, 1]$ |
| Hapke | $x = \dfrac{w}{\left(1 + 2\mu\sqrt{1-w}\right)\left(1 + 2\mu_0\sqrt{1-w}\right)}$ <br> $\sqrt{1-w} = \frac{\sqrt{(\mu_0 + \mu)^2 x^2 + (1 + 4\mu_0 \mu x)(1-x)} - (\mu_0 + \mu)x}{1 + 4\mu_0 \mu x}$ | $\mu_0$: cosine incident angle <br> $\mu$: cosine reflectance angle |

### 4.1.2. Comparison to other Mapping Methods

We will compare the proposed method to methods that learn a mapping onto the Fan model and the Hapke model respectively and refer to these as GP_Fan, KRR_Fan, NN_Fan and GP_Hapke, KRR_Hapke, and NN_Hapke repectively. Inversion of the Fan model and the Hapke model then reveal the fractional abundances of the test spectra.

We also compare to the direct mapping strategy from the literature, in which a mapping is learned to the fractional abundances. We will apply the same mapping methods as for the proposed approach and simply refer to these methods as GP, KRR, and NN. We also applied a support vector regression (SVR) to perform direct mapping. Finally, we would like to point out that when using the NN method, the nonnegativity and sum-to-one constraints can be enforced, by applying a softmax activation function in the last layer of the network. We applied this method as well in the comparison and refer to it as SM.

### 4.1.3. Performance Measures

The performance of the different methods was assessed using the root mean squared error (RMSE) between the estimated and ground truth fractional abundances:

$$\text{RMSE}\,(\%) = \sqrt{\frac{1}{(N-n)p} \sum_{j=1}^{p} \sum_{i=n+1}^{N} (a_{\mathrm{e}_{ji}} - \hat{a}_{\mathrm{e}_{ji}})^2} \times 100, \tag{18}$$

where $a_{\mathrm{e}_{ji}}$ is the true fractional abundance of endmember $j$ for pixel $i$, $\hat{a}_{\mathrm{e}_{ji}}$ is the estimated one, $N - n$ is the number of test samples, and $p$ is the number of endmembers.

The mapping accuracy of the different methods was assessed using the reconstruction error (RE) between the mapped test spectra and the unmixing results.

$$\text{RE} = \sqrt{\frac{1}{(N-n)d} \sum_{i=n+1}^{N} \|\mathbf{x}_i - \hat{\mathbf{x}}_i\|^2} \times 10000 \tag{19}$$

where $\hat{\mathbf{x}}_i$ is the reconstructed spectrum and $d$ is the number of wave-bands.

*4.2. Experiments Using the Simulated Mineral Dataset*

4.2.1. Comparison with Direct Mapping to the Fractional Abundances

In this experiment, 20 realizations of 10,000 mixed spectra were generated by applying the Hapke model, each time using three randomly selected mineral endmembers from the USGS library. A small amount of Gaussian noise (signal-to-noise ratio (SNR) = 50 dB) was added. To learn the map, two different training sets were applied; in the first experiment, the endmembers themselves were applied as training samples, as was done by some of the works in the state of the art. In the second experiment 10 randomly selected mixed spectra were applied as training samples. For NN_LM, NN, and SM, the training samples were further subdivided into a training and a validation subset. Only the training subset was used to estimate the network parameters. GP_LM and GP used all training samples to optimize the hyperparameters, while the hyperparameters involved in KRR_LM and KRR were optimized by 10-fold cross-validation of the training samples.

In Table 2, we show the results on this dataset, using GP_LM, KRR_LM, and NN_LM, by mapping onto the linear model, their counterparts GP, KRR, and NN and SVR and SM by mapping directly onto the fractional abundances. The table shows the RMSE and the fraction of spectra for which negative fractional abundances were estimated (NEFA) (%).

**Table 2.** Results on 20 runs of 10,000 test pixels generated by the Hapke model. The best performances are denoted in bold.

| Method | GP_LM | GP | KRR_LM | KRR | NN_LM | NN | SVR | SM |
|---|---|---|---|---|---|---|---|---|
| | | | | training set 1 | | | | |
| RMSE | **19.88** ± 0.62 | 40.89 ± 0.01 | 31.81 ± 1.71 | 40.89 ± 0.01 | 23.57 ± 1.97 | 36.57 ± 6.40 | 34.91 ± 9.48 | 33.68 ± 0.01 |
| NEFA | 0 | 48.32 ± 0.31 | 0 | 25.12 ± 0.27 | 0 | 22.34 ± 0.36 | 24.37 ± 0.62 | 0 |
| | | | | training set 2 | | | | |
| RMSE | **3.05** ± 1.10 | 5.54 ± 1.31 | 4.05 ± 0.58 | 5.55 ± 0.95 | 4.15 ± 1.17 | 5.15 ± 0.80 | 7.10 ± 0.95 | 15.65 ± 5.88 |
| NEFA | 0 | 5.13 ± 1.97 | 0 | 4.66 ± 0.82 | 0 | 8.71 ± 2.40 | 8.96 ± 1.67 | 0 |

We can conclude that in both experiments, the mapping onto the linear model outperformed the direct mapping onto the abundances. In the latter cases, a substantial fraction of the estimated abundances had negative values. Remark that when the training set of pure endmembers was applied, no method was able to do better than the linear model (RMSE of 18.64 ± 1.07), since no nonlinearity was contained in the training set. Since SM is the only direct mapping method that is constrained, we will apply it for comparison in the remaining experiments.

4.2.2. Robustness to Noise and the Number of Endmembers

To test the proposed strategy for its robustness to noise, three endmembers were randomly selected from the USGS spectral library. Mixed spectra were produced by using the five different spectral mixing models: LMM, the Fan model [4], PPNM [4], MLM, and the Hapke model. A total of 500 spectra (100 spectra for each mixing model) were generated. After generating the spectra, different levels (10, 20, 30, 40, 50 dB) of Gaussian noise were added. 250 spectra were applied as training samples and testing was performed on the remaining 250 samples. Figure 6 plots the obtained average and standard deviation of the RMSE from the supervised methods as a function of noise level from 20 independent runs.

For the proposed methods, reasonably low RMSE values were obtained, showing that they were able to learn nonlinearities of different nature simultaneously for different levels of noise. GP_LM clearly outperformed NN_LM, KRR_LM, and SM.

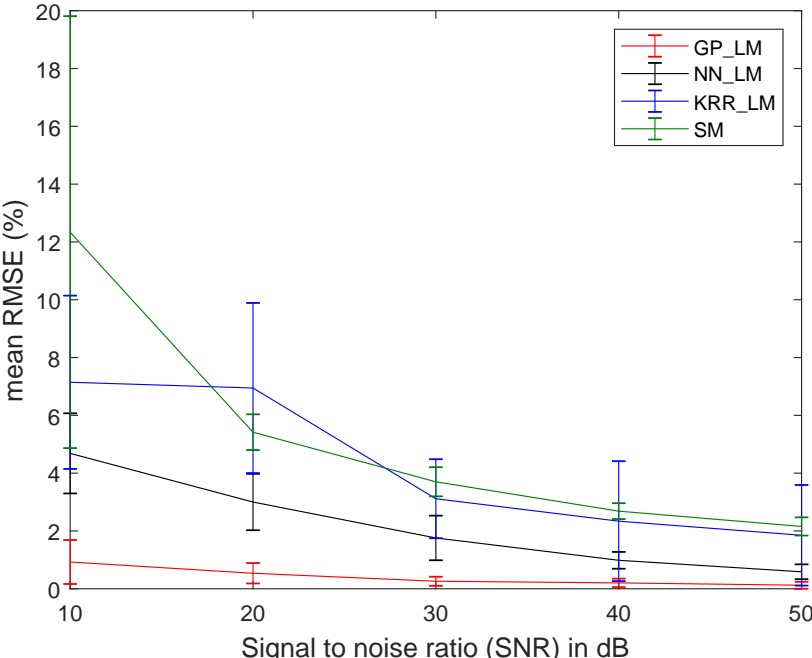

**Figure 6.** Root mean squared error (RMSE) (20 runs) for the methods using the proposed strategy (Gaussian processes (GP)_LM, kernel ridge regression (KRR)_LM, and neural networks (NN)_LM) and neural networks method with the softmax activation function in the last layer of the network (SM) as a function of signal-to-noise ratio (SNR).

To test the robustness to the number of endmembers, different numbers of endmembers (3–10) were used to generate 500 mixed spectra by using the five mixing models mentioned above. After generating the spectra, 30 dB Gaussian noise was added. Half of the samples were used for training and a half for testing. Figure 7 plots the obtained average and standard deviation of the RMSE from the supervised methods as a function of the number of endmembers from 20 independent runs.

Again, the methods using the proposed strategy performed better than SM, and GP_LM outperformed the others. Even for higher numbers of endmembers, the obtained RMSE values remained low.

4.2.3. Accuracy of the Mapping

An experiment was performed on the data from Section 4.2.2, with 30 dB of Gaussian noise and three mineral endmembers, to assess the accuracy of the different mappings. Table 3 lists the RMSE and RE for 20 realizations, when mapping onto the linear model, the Fan model and the Hapke model respectively. The results demonstrate that mapping to the linear model produces the lowest RMSE and RE.

**Table 3.** Root mean squared error (RMSE) and reconstruction error (RE) (20 runs) of 250 test pixels of the simulated dataset. The best performances are denoted in bold.

| Method | GP_LM | GP_Fan | GP_Hapke | KRR_LM | KRR_Fan | KRR_Hapke | NN_LM | NN_Fan | NN_Hapke |
|--------|-------|--------|----------|--------|---------|-----------|-------|--------|----------|
| RMSE | **1.19** $\pm$ 0.72 | 1.44 $\pm$ 0.93 | 1.46 $\pm$ 0.74 | 3.04 $\pm$ 0.32 | 3.06 $\pm$ 0.35 | 3.61 $\pm$ 0.64 | 3.65 $\pm$ 0.59 | 4.12 $\pm$ 0.52 | 5.20 $\pm$ 1.25 |
| RE | **2.4** $\pm$ 1.98 | 5.8 $\pm$ 4.00 | 19 $\pm$ 18 | 6.25 $\pm$ 3.28 | 12 $\pm$ 2.14 | 21 $\pm$ 4.47 | 15 $\pm$ 7.37 | 29 $\pm$ 7.32 | 50 $\pm$ 22 |

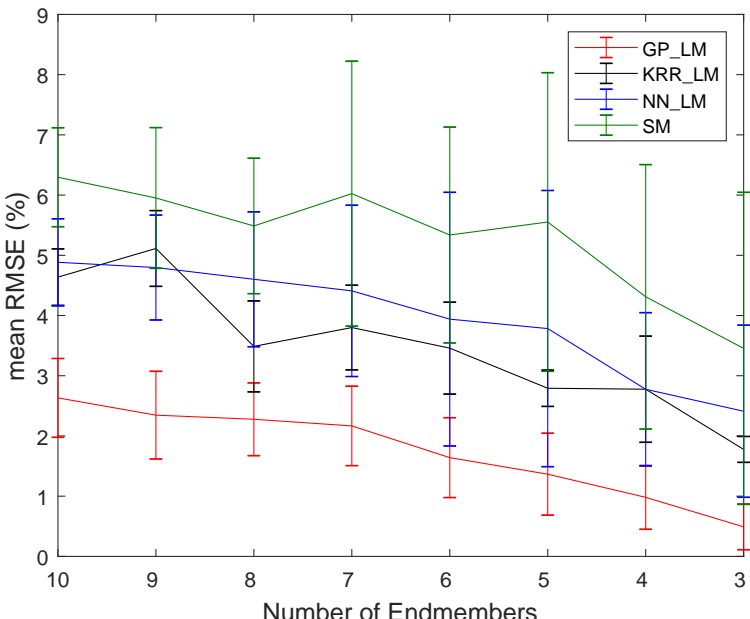

**Figure 7.** RMSE (20 runs) for the methods using the proposed strategy (GP_LM, KRR_LM, and NN_LM) and SM as a function of the number of endmembers.

### 4.3. Experiments on the Ray Tracing Vegetation Dataset

The entire ground truth data set was divided into a randomly selected training and test set. Figure 8 plots the obtained average and standard deviation of the RMSE from the supervised methods of the proposed strategy and SM as a function of the applied number of training samples from 100 independent runs. For all cases, the error was reduced when the number of training samples was increased. GP_LM performed the best.

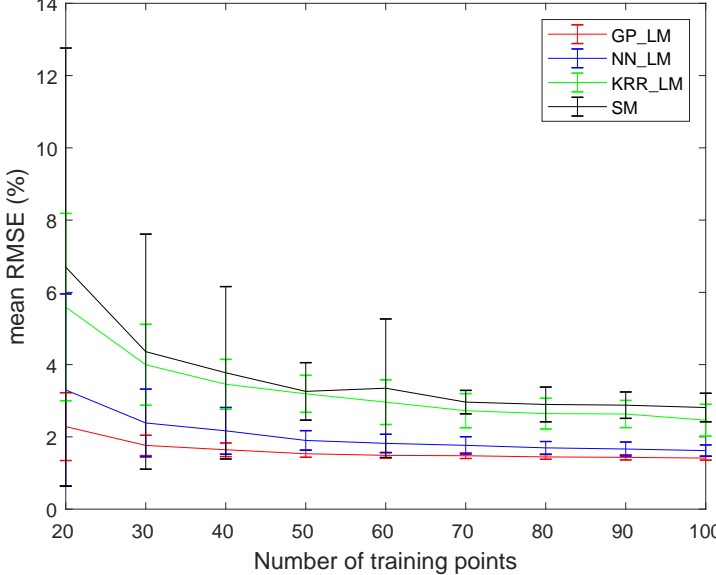

**Figure 8.** RMSE (100 runs) obtained by the four studied supervised methods as a function of the applied number of training samples in the ray tracing vegetation dataset.

Table 4 lists the obtained results by all methods for 100 experiments (40 training and 360 test samples). The table shows the obtained RMSE between the estimated and ground truth fractional abundances for each endmember (in Equation (18) *p* was replaced by one). Among all supervised methods, GP_LM outperformed the others. When unsupervised techniques were applied, the errors were much larger than when using the proposed approach.

**Table 4.** RMSE (100 runs) of 360 test pixels of the ray tracing dataset. The best performances are denoted in bold.

| Endmember Method | GP_LM | KRR_LM | NN_LM | SM | LMM | Fan | PPNM | MLM | Hapke |
|---|---|---|---|---|---|---|---|---|---|
| Soil | **1.59** ± 0.16 | 2.39 ± 0.93 | 1.86 ± 0.34 | 3.07 ± 0.46 | 11.69 | 12.40 | 16.15 | 14.06 | 9.79 |
| Weed | **1.18** ± 0.18 | 3.77 ± 1.06 | 1.62 ± 0.40 | 3.86 ± 1.42 | 14.84 | 12.72 | 21.40 | 14.42 | 5.94 |
| Tree | **2.06** ± 0.21 | 3.95 ± 0.67 | 2.63 ± 0.46 | 3.74 ± 1.20 | 24.36 | 19.13 | 8.28 | 26.06 | 13.39 |

Figure 9 displays (for one experiment) the ground truth (GT) and estimated abundance maps, and the absolute difference between the GT and the estimated ones for the proposed supervised methods. All images are normalized by the largest abundance value in the GT map. The grayscale in the figures ranges from zero (black) to one (white). There was no significant difference between the abundance maps obtained by GP_LM and NN_LM. Compared to GP_LM and NN_LM, KRR_LM and SM showed a low performance for the estimation of weed.

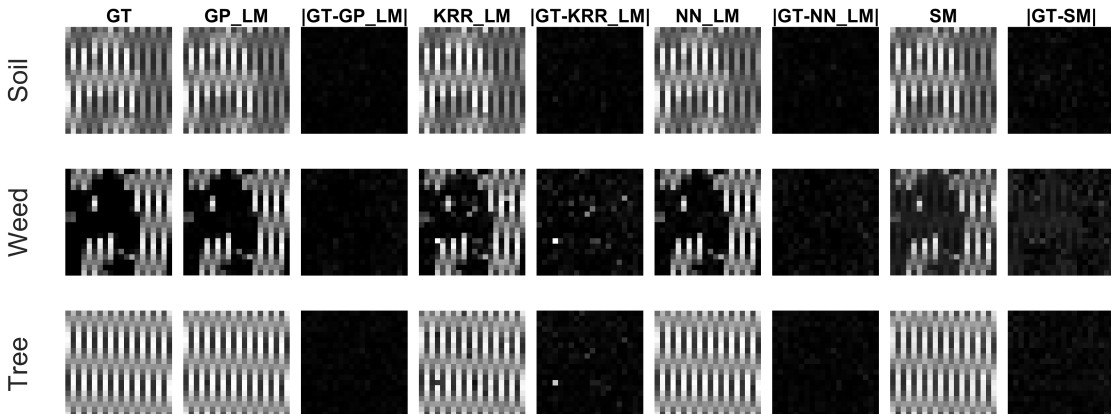

**Figure 9.** Ground truth (GT) and estimated abundance maps and absolute difference with the GT for the four proposed supervised methods on the ray tracing dataset. All images are normalized by the largest abundance value in the GT map.

## 4.4. Experiments on the Drill Core Hyperspectral Dataset

The entire ground truth data set was divided into a (randomly selected) training and test set. Figure 10 plots the obtained average and standard deviation of the RMSE for 100 runs from the studied methods as a function of the applied number of training samples that were selected randomly. These results indicate that the error is reduced when increasing the number of training samples. KRR_LM performed slightly better than GP_LM, NN_LM, and SM.

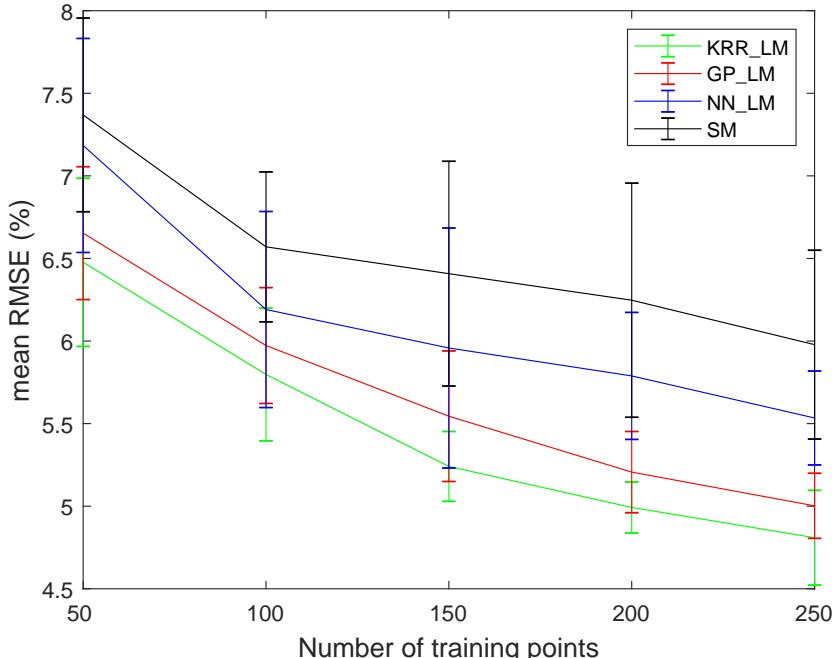

**Figure 10.** RMSE (100 runs) obtained by the four studied supervised methods as a function of the applied number of training samples in the drill core dataset.

In Table 5, we show the obtained results of the supervised methods for 100 experiments with 187 training and 186 test samples. The table shows the estimated fractional abundances, averaged over all test samples. Here, GT is the mean fractional abundance over the entire MLA region, i.e., of all ground truth (373) pixels. Overall, the estimated abundances were close to the ground truth, even for the minerals with very low fractional abundances. KRR_LM performed better for the estimation of Pyrite. GP_LM outperformed the others for the estimation of Chlorite and Barite. NN_LM performed best for the estimation of White Mica, Biotite, Amphiboles, Gypsum, K-Feldspar, and Quartz. Remark however that the standard deviations of NN_LM were much larger than for GP_LM and KRR_LM. The performance of SM was low for the estimation of low abundant minerals. SM, however, outperformed all other techniques for the estimation of Plagioclase, i.e., the most abundant mineral. From these results, we can conclude that mapping directly to the fractional abundances is suitable only for high abundant minerals.

Table 5 also shows the mean estimated fractional abundances of all 373 pixels when performing the five different unsupervised spectral unmixing methods (here, endmembers were the same as in the supervised experiments, i.e., obtained from the USGS spectral library). Most of the unsupervised techniques did not perform well on any of the minerals. One possible reason for the low performance of the mixing models is that they heavily rely on the endmember spectra obtained from the USGS spectral library, which are not corresponding to the actual endmembers in the data. Since pure materials are not available in the mineral mixtures, one way to avoid this endmember variability would be to capture the spectra of the pure minerals along with the acquisition of the hyperspectral image. But even then, the nonlinearities of the data will never correspond entirely with any of the applied models. The proposed supervised strategy seems to better account for both the nonlinearities and endmember variabilities through the mapping.

**Table 5.** The mean estimated fractional abundances of 186 test pixels (in %) of the drill core dataset using the supervised methods (100 runs) along with the estimated fractional abundances using the unsupervised methods. The best performances are denoted in bold.

| Mineral Method | GT | KRR_LM | GP_LM | NN_LM | SM | LMM | Fan | PPNM | MLM | Hapke |
|---|---|---|---|---|---|---|---|---|---|---|
| White Mica | 14.14 | 13.40 ± 0.83 | 13.57 ± 0.79 | **13.61** ± 1.22 | 13.56 ± 1.32 | 9.01 | 3.82 | 6.93 | 9.63 | 6.91 |
| Biotite | 0.46 | 0.63 ± 0.12 | 0.62 ± 0.12 | **0.59** ± 0.20 | 1.04 ± 0.61 | 0 | 0 | 0 | 0 | 0 |
| Chlorite | 4.17 | 4.66 ± 0.35 | **4.56** ± 0.33 | 4.61 ± 0.85 | 3.64 ± 0.60 | 0 | 0 | 0 | 0 | 0 |
| Amphiboles | 1.43 | 1.37 ± 0.10 | 1.38 ± 0.09 | **1.43** ± 0.40 | 1.83 ± 0.46 | 0.24 | 0 | 6.46 | 0.81 | 0 |
| Gypsum | 10.93 | 11.21 ± 1.24 | 11.11 ± 1.19 | **11.08** ± 1.20 | 11.71 ± 1.61 | 32.32 | 31.31 | 44.26 | 31.66 | 42.54 |
| Plagioclase | 33.20 | 32.24 ± 1.58 | 32.74 ± 1.44 | 32.32 ± 2.69 | **33.72** ± 2.69 | 7.30 | 3.66 | 7.62 | 20.43 | 0 |
| K-Feldspar | 6.68 | 5.93 ± 0.30 | 5.98 ± 0.32 | **6.27** ± 1.28 | 5.58 ± 0.76 | 4.77 | **6.33** | 4.92 | 7.74 | 41.21 |
| Quartz | 26.85 | 28.15 ± 1.26 | 27.62 ± 1.12 | **27.19** ± 2.52 | 25.91 ± 1.70 | 0.43 | 0.14 | **20.64** | 3.79 | 0 |
| Barite | 0.33 | 0.46 ± 0.10 | **0.43** ± 0.09 | 0.45 ± 0.24 | 0.98 ± 0.59 | 0 | 0 | 3.40 | 0 | 0 |
| Pyrite | 1.80 | **1.95** ± 0.32 | 2.00 ± 0.34 | 2.46 ± 1.14 | 2.02 ± 0.44 | 45.93 | 54.73 | 5.76 | 25.93 | 9.34 |

Figure 11 shows the ground truth abundance maps (GT), the estimated abundance maps, and the absolute difference with the ground truth map, obtained by GP_LM, KRR_LM, NN_LM, and SM respectively for one experiment (all images are rescaled to the maximal GT abundance of each mineral). GP_LM and KRR_LM perform well on abundantly present minerals but have a low performance on the estimation of minerals with low abundance, such as Biotite, Barite and Pyrite. The performance of NN_LM is generally lower than GP_LM and KRR_LM. The obtained abundance maps are blurred. SM shows a low performance for the estimation of almost all minerals, except for Plagioclase, the most abundant mineral.

To validate the methods, they were applied to the entire drill core sample, when the mapping is learned from the MLA region (red rectangle in Figure 4a). One problem to overcome is the spectral variation along the sample. A scatterplot on three spectral bands is shown in Figure 12a. From this figure, it is quite clear that the ground truth spectra (represented by red circles) occupy only a small portion of the spectral space covered by the entire drill core sample (represented by blue points). As a consequence, the learned map will not be representative for the entire sample.

The solution to the problem is to normalize the dataset before applying the model. For this, each spectrum is divided by its norm. The obtained scatterplot after normalization is shown in Figure 12b. After normalization, the abundance maps of the whole drill core sample (see Figure 4a) were estimated by using all 373 ground truth samples as the training set. Figure 13 shows the abundance maps estimated by GP_LM, NN_LM, and KRR_LM respectively for the entire drill core sample.

From Figure 13 it can be observed that for most of the minerals, these results give a general representation of their expected distribution and abundance in the drill core sample. Quartz, gypsum, and pyrite have been mainly mapped in the veins of the drill core sample. However, pyrite content has been overestimated (i.e., the estimated fractional abundance values are higher than expected) by the three methods. This can be attributed to the featureless nature of both pyrite and quartz minerals in the SWIR region of the electromagnetic spectrum. Although white mica content seems to be underestimated (i.e., the estimated fractional abundance values are lower than expected), its major content has been correctly mapped in the alteration halos. Chlorite and a small amount of quartz were also mapped in the alteration halos. The matrix of the drill core sample has mainly been occupied by the three methods with k-feldspar, plagioclase, and less biotite. However, biotite content is being overestimated in the central vein.

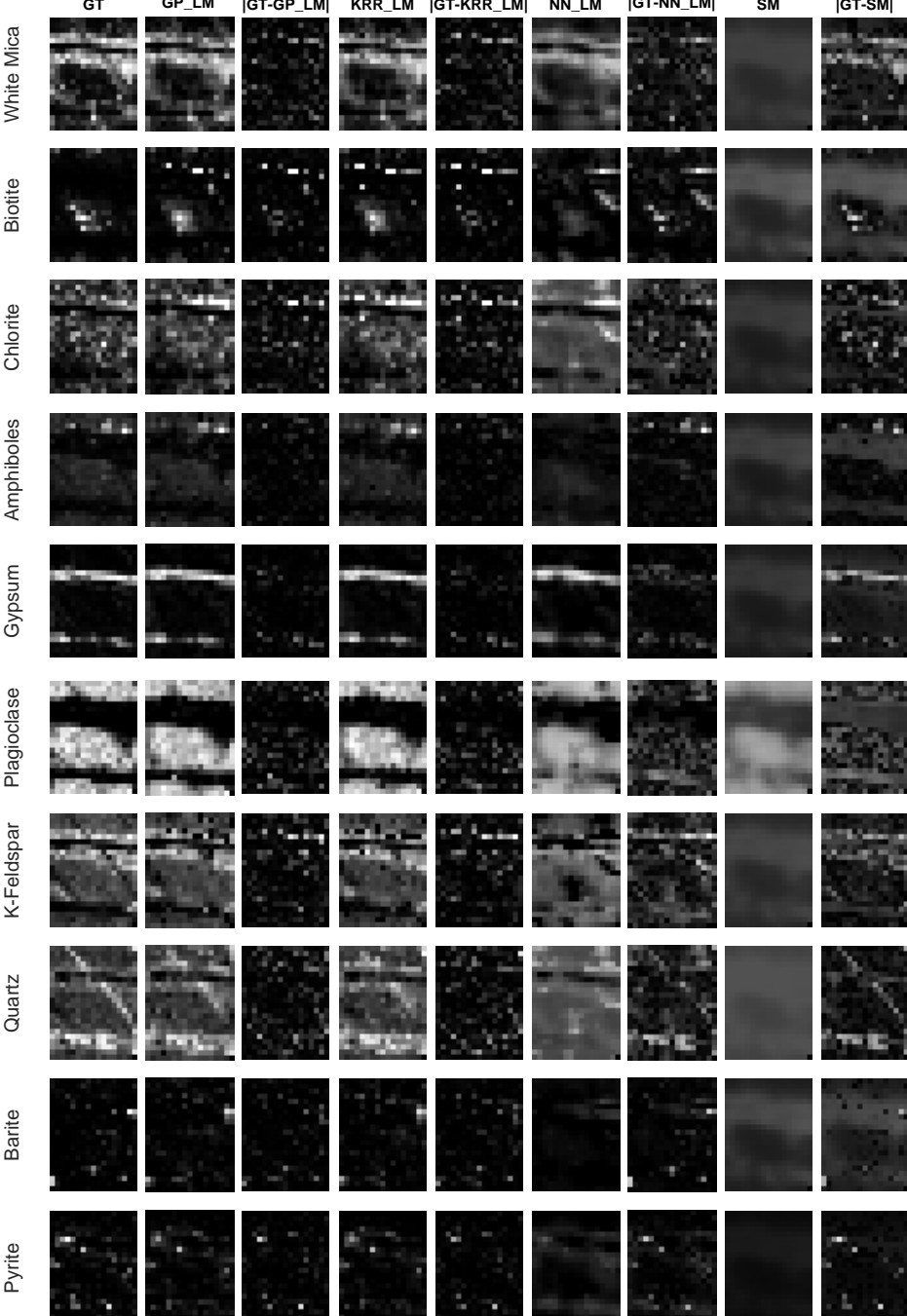

**Figure 11.** Estimated abundance maps and absolute differences with the ground truth for the four supervised methods on the drill core dataset. All images are normalized by the largest abundance value in the GT map.

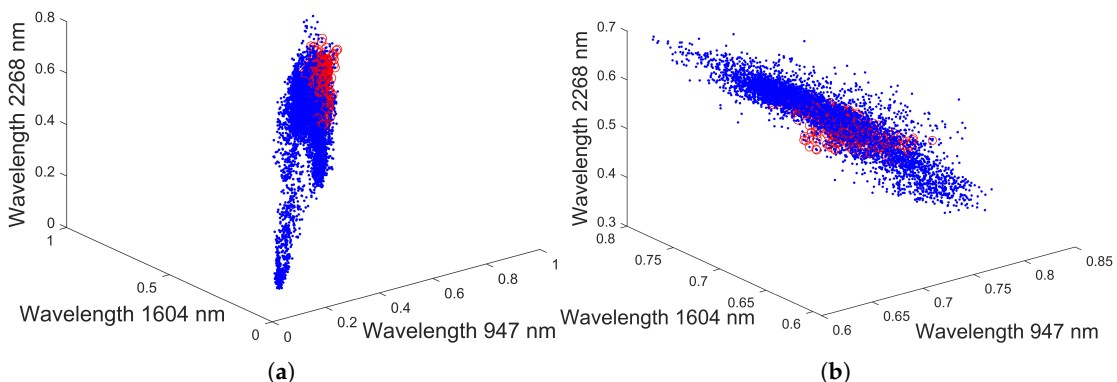

**Figure 12.** Scatterplot of the whole drill core sample (blue dots) and the MLA region (red circles); without (**a**) and with (**b**) normalization.

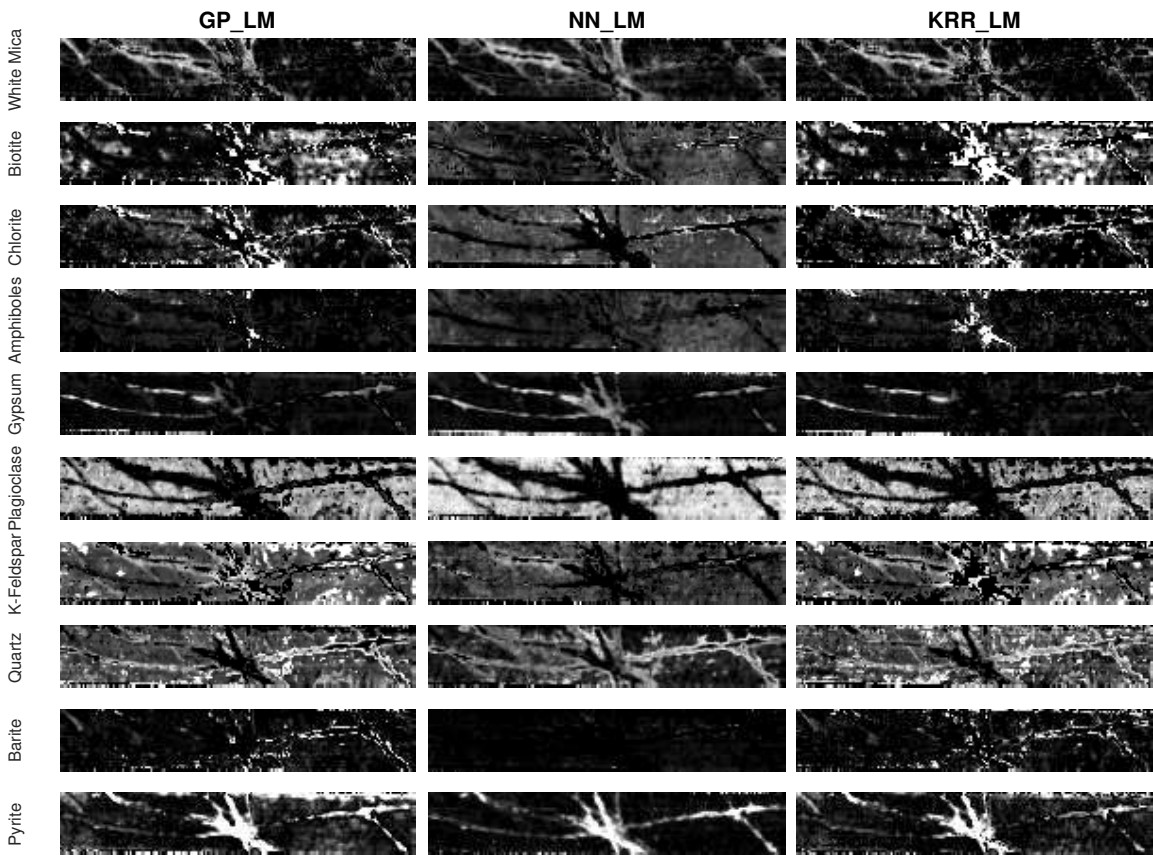

**Figure 13.** Estimated abundance maps of entire drill core sample (applying a map learned by using 373 training pixels). All images are normalized by the largest abundance value in the GT map.

Figure 14 shows the estimated abundance maps (without rescaling to the maximal abundance of each mineral) of minerals, grouped according to their occurrence in the veins and the matrix. Although GP_LM and KRR_LM performed better in estimating the minerals in the ground reference MLA region (see. Figures 10 and 11), NN_LM seems to outperform the other two methods in distinguishing between veins and matrix in the entire drill core sample. A possible explanation is that NN_LM is more capable of generalization while GP_LM and KRR_LM may overfit the training samples.

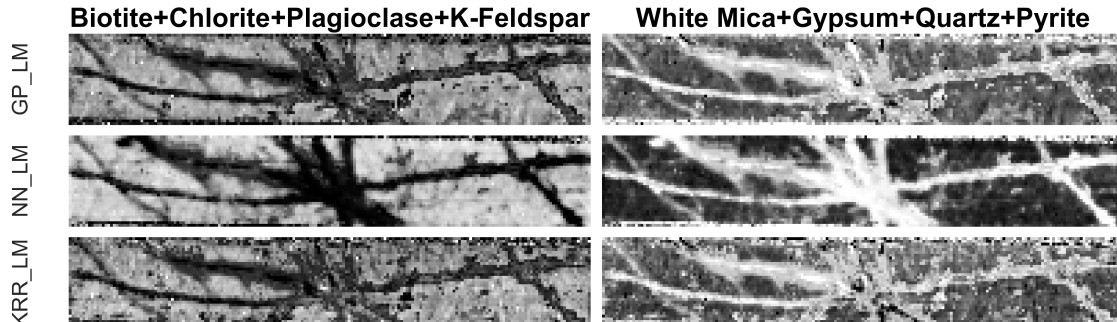

**Figure 14.** Estimated abundance maps (without rescaling) of the entire drill core sample, for groups of minerals representing the matrix (**left**) and veins (**right**) (applying a map learned by using 373 training pixels).

## 5. Discussion

From the experiments, the following general conclusions can be drawn:

- The supervised methods all outperform the use of nonlinear spectral mixture models. This is of course partially due the fact that these methods make use of training data. However, the fact that they do not rely on a specific mixture model and the generic nature of these methods allows them to better account for nonlinearities and spectral variability in the data. Results show that the supervised methods can take nonlinearities of different nature simultaneously into account.

- The strategy of mapping onto the linear mixture model outperforms methods that directly map onto the fractional abundances. The main difference is that the proposed methodololgy inherently meets the nonnegativity and sum-to-one constraints. As opposed to the direct unconstrained mapping of the spectra to abundances, the estimated fractional abundances have a clear physical meaning and consequently the estimates are more accurate. Another difference with direct mapping is that our approach requires endmembers, while the direct mapping methods do not. A clear advantage is that in case no pure pixels are available in the data, endmember spectra from a spectral library can be applied. The mapping accounts for the spectral variability between these and the actual endmembers of the data.

- The proposed methodology is generic in the sense that a mapping can be learned to any model. However, learning a mapping to the linear model is favorable over learning mappings to nonlinear models. The higher the nonlinearity of the model, the higher the errors seem to be. The reason for this is that it is just easier to project onto a linear manifold, since a linear manifold can more easily be characterized by a limited number of training samples.

- The proposed methodology requires high quality ground truth data. Learning the mapping based on pure or linearly mixed spectra, as is done in the state of the art literature, will not improve results over the linear mixture model. High-quality ground truth data for spectral unmixing is hard to obtain, in particular in remote sensing applications. One way is to make use of a high-resolution multispectral image of the same scene (if available) to generate a groundcover classification map that can be used to generate ground truth fractional abundances for a low-resolution hyperspectral image. However, in the domain of mineralogy, it is more common to generate validation data with other characterization techniques, such as the MLA technique in the drill core example.

- An advantage of the proposed methodology is that any nonlinear regression method can be applied for learning the mapping. In this work, we compared three different ones. Gaussian processes generally seems to outperform kernel ridge regression and feedforward neural networks. Compared to kernel ridge regression, Gaussian processes contains more hyperparameters for a band-by-band adaptation to the nonlinearities. However, Gaussian processes can overfit the data in case the ground truth fractional abundance values are not very trustworthy. This can be observed in the obtained abundance maps of the entire drill core sample (Figures 13 and 14).

On the other hand, a neural network has better generalization properties, but its training can be computationally expensive.

All methods were developed in Matlab and ran on an Intel Core *i7*-8700*K* CPU, 3.20 GHz machine with 6 cores. The runtimes of the proposed methods on the ray tracing vegetation dataset and the drill core dataset are shown in Table 6. As can be seen, the runtime of NN_LM is relatively high for the hyperspectral drill core mineral dataset due to the involvement of a large number of network parameters (approx. 8400). KRR_LM has the lowest runtime because it involves only two free parameters. In GP_LM, 187 and 402 parameters needed to be estimated for the ray tracing vegetation dataset and the drill core dataset respectively.

**Table 6.** The runtime in seconds.

| Method | time$_{\text{Ray tracing}}$ (s) | time$_{\text{Drill core}}$ (s) |
|--------|------------------|----------------|
| KRR_LM | 0.89 | 20.55 |
| GP_LM | 43.84 | 840.62 |
| NN_LM | 105.15 | 7499.84 |

## 6. Conclusions

In this paper, we proposed a supervised methodology to estimate fractional abundance maps from hyperspectral images. The method learns a map of the actual spectra to the corresponding linear spectra, composed of the same fractional abundances. Three different mapping procedures, based on artificial neural networks, Gaussian processes, and kernel ridge regression were proposed, but in principle, any nonlinear regression technique can be applied. The methods were validated on a ray-tracing vegetation dataset, a simulated, and a real mineral dataset.

The proposed methodology is superior to unsupervised nonlinear models, which in most cases produced large errors or gave unacceptable results. The methodology also outperforms a direct mapping of the spectra to the fractional abundances. On the vegetation dataset and the simulated mineral dataset, Gaussian processes performed the best. From the experiments on the real drill core dataset, we can conclude that the mapping does not only account for nonlinear effects but also allows the use of library endmembers for the estimation of the fractional abundances. The mapping method based on artificial neural networks provided the best abundance maps of the real drill core sample.

To even better account for endmember variability, the proposed approach can be extended by assigning more than one fixed endmember to each pure material, and learning a mapping to different linear manifolds, after which MESMA can be applied. This will allow us to estimate the endmembers and the corresponding fractional abundances of each pixel. In the future, we will apply the proposed methodology to applications where high-quality ground truth data is more readily available, such as in chemical mixtures and in mixed and compound materials.

**Author Contributions:** Conceptualization, B.K., Z.Z. and P.S.; methodology, B.K. and P.S.; software: B.K.; validation, B.K., M.K, C.C and R.G.; writing—original draft preparation, B.K.; writing—review and editing, M.K., Z.Z., C.C., R.G. and P.S.

**Funding:** This research was funded by BELSPO (Belgian Federal Science Policy Office) in the frame of the STEREO III programme—project GEOMIX (SR/06/357).

**Conflicts of Interest:** The authors declare no conflict of interest.

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
