# Peer review of "A Supervised Method for Nonlinear Hyperspectral Unmixing"

_remotesensing, doi:10.3390/rs11202458_

Round 1
Reviewer 1 Report
I Accept the article in present form.
Reviewer 2 Report
The authors have addressed this reviewers comments. I recommend this paper for publication. Please check some grammatical errors.
Reviewer 3 Report
This paper proposes a supervised method for nonlinear hyperspectral unmixing. The idea of learning a nonlinear mapping to map nonlinear spectra to linearly mixed spectra is interesting and makes sense. The paper is well organized and easy to understand. However, the authors are suggested to take the following comments to further improve the quality of this paper.
The idea of learning a nonlinear mapping is quite similar to the kernel based unmixing methods, which also turn nonlinear problems into linear ones through the nonlinear mapping implicitly determined by the kernel function. In particular, the mapping using KRR is similar to KFCLS. The authors are suggested to make comparison with the kernel based methods such as KFCLS, KNMF and add some discussion on this.
In 4.2.2, five different spectral mixing models including LMM, the Fan model [4], PPNM [4], MLM, and the Hapke model are adopted to generate mixed spectra. But in 4.2.1, only the Hapke model is used and the scale of mixed spectra is much larger than that of 4.2.2. Please explain the reason.
Once the mapping is learned, LMM model can be applied to the mapped test spectra and the fractional abundances can obtained. Therefore, the reconstruction error between the mapped test spectra and the unmixing results is suggested to be calculated and presented, which can demonstrate the performance of the proposed method. Moreover, it would be interesting to investigate the RE performance of the proposed method on different mixing models, which can also validate the effect of learning a mapping.
Reviewer 4 Report
This paper proposed a supervised method for nonlinear spectral unmixing. Three methods are presented for learning the nonlinear mapping, based on feed-forward neural networks, kernel ridge regression, and gaussian process. Experimental results conducted on artificial and real hyperspectral data sets. The idea generally makes sense and the results look encouraging. However, the presentation is not in good style, and there are still some problems to be addressed before its publication in remote sensing. Please refer to the following comments.
The logic of presentation, especially in the Introduction is weak. In formula (7) and(15), the specific map learned by KRR and GP is presented, which mapping the actual training spectra to the generated linear spectra. But there is no specific map or function in NN method. Although it is researched in [33], it is necessary to show how to map the actual training spectra to the generated linear spectra. The loss function, the significant component in NN, should be presented in line 139, where the structure and activate function of the network are shown. The word “map” occurred for many times (line 8, 91, 92, 108, 124, 128-130, 132) but for a different meaning, it should be declared that which map is “direct map” or “just a map to linear spectra”. The parameters needed to be trained in KRR and GP are referred in line 147 and 154 respectively. While there aren’t weights needed to be learned in the NN method. The way of division of the training data, validation data and test data should be clearly written (line 257 266 273 278 289 294-295)
7. The number of the endmembers and the fractional abundance of drill core hyperspectral dataset should be presented clearly in line 304. Otherwise, the readers will be confused
8. In line 382 and 383, maybe you can use the state of the art framework in deep learning to train NN to accelerate the process of training by using GPU.
Considering that the existence of noise is inevitable in hyperspectral image and the setting of the noise in the latter experiments, a statement should be added in line 100, which states that the generalization of the mapping procedure allows to learn and avoid the information of noise simultaneously. Formula 17 is refer to the test dataset, so the index i should range from n+1 to N instead of 1 to N-n. The learned map can map the real nonlinear spectra to linearly mixed spectra, but the essence of the process of unmixing is still using a linear model. Can the drawback of the linear model mentioned in introduction be avoided? The method NN using softmax as activate function (line 231 249) can also be as an indirect map. Why does it not appear in latter experiments? In the experiments on the simulated mineral dataset, the difference is compared between the method proposed and the direct mapping. Why this comparison disappear in the experiments on the ray tracing vegetation dataset and the drill core hyperspectral dataset? There are some concepts mentioned few times are not clear in this paper: a) what does “absolute difference” mentioned in line 284 and 318 mean? In the image, the more black means the absolute difference larger? b) what does “unpolished” in line 194 and 200 mean? c) what does “spatial resolution” in line 50, 188 and 199 mean? d) what does “high resolution” in line 203 and 204 mean? e) what does “overestimate and underestimate” in line 339 and 341 mean? f) what does “high uncertainty” in line 380 mean?
Please give more clear explanations to the above question.
Round 2
Reviewer 3 Report
The authors have answered all my questions. There is no more comment.
Reviewer 4 Report
Almost all my concerns have been solved. I'd like to see it to be accepted.
This manuscript is a resubmission of an earlier submission. The following is a list of the peer review reports and author responses from that submission.
Round 1
Reviewer 1 Report
In the article, authors proposed a supervised methodology to estimate fractional abundance maps from hyperspectral images.
Unfortunately, the research has been presented in a chaotic manner. The basic principle of separating the description of individual components of the article, ie a clear division into sections: description of data, methodology, description of results, discussions and summary, has not been preserved. All of these sections have been mixed up and, therefore, the article loses greatly.
In addition, the Introduction contains a too poor review of the literature on the Spectral Unmixing issue raised, especially when it comes to papers from the last 5 -10 years - please complete and expand the introduction part.
Moreover:
1. Suggesting a precise separation of the description of experimental data to one section (without mixing them in the results of individual experiments). Why did the authors, in order to compare the operation of their method, not use in the experiment real data recorded with a hyperspectral camera for the real area where the examined objects occur?
2. The description of the methodology, which is currently in section 2 and in section 3 "Experimental results and disussion", should be prepared in one section with a clear division into the description of the methods used and the exact implementation of the next steps performed during the experiments
3. Section 3 has been denounced "Experimental results and discussion", but unfortunately the authors do not discuss with any results known from the literature
4. For clarity and easier reading of the article, please insert figures 9, 10, 11, 12 in the text of the relevant sections, and not at the end of the article for the literature section
Reviewer 2 Report
This paper compares different methods for nonlinear hyperspectral unmixing. Specifically, this paper compares methods that use training data and learn a map between linearly mixed spectra and actual spectra. Overall, the manuscript is well-written. However, this paper needs to address several comments before this paper is accepted for publication.
Major comments
1. Is it realistic to assume that actual spectra and the fractional abundances are available in real data? I’d like to know how the authors prepare for the ground truth abundance maps in more details. Can the ground truth abundance maps be prepared for other datasets, for example, Cuprite data? Or the authors design this experiment only for the mineral dataset?
2. I’m aware that there are many studies that directly learn a map from actual spectra to abundances. For example, [R1] used NN to learn the map. [R2] used SVR to learn the map. [R3] used GP. These methods also face the problem of acquiring training data (fractional abundances). [R2] synthetically generates the mixed spectra and their corresponding abundances and use them as training data. [R3] assumes the linear unmixing and avoid using fractional abundances of mixed spectra. The proposed method of this paper is different to the above methods because it tries to learn a map from actual spectra to linearly mixed spectra. As the authors stated, it may be difficult to impose the sum-to-one constraint if a map from actual spectra to abundances is learned. However, it is still not clear how the proposed method is different to existing methods in terms of the acquisition of training data or performance. This problem comes from the insufficient literature review and lack of comparison. I suggest the two points.
- the authors review existing methods that use training data for hyperspectral unmixing and discuss the problem of existing methods (especially acquisition of training data) and how the proposed method works better than existing methods.
- the authors do a simple experiment to compare the existing approach that learns a map from actual spectra to abundances and the proposed approach. Experiment on one synthetic data should be sufficient.
3. When comparing methods, it is good to compare them in a fair setting. In the paper, KRR used RBF for the kernel while GP used squared exponential kernel which is more flexible. I would suggest that the authors use a same kernel for both methods. And the authors need to discuss how the parameters of KRR are determined.
Minor comments
1. This paper focuses on the comparison of different supervised methods rather than developing a method. I suggest that the title of this paper is changed to include the word “comparison”.
2. In Figure 1, “Training hyperspectral image (HSI)” is a bit confusing. I think that the authors do not use the whole image but use actual spectra from HSI?
3. In the experimental results, the authors simply state that GP outperformed the others or similar sentences. I suggest that the authors discuss why GP outperformed the others.
4. Results are not consistent for simulated mineral data and real data. While GP outperforms other methods in simulated mineral data, NN outperforms other methods in real mineral data. If an optimal method is changed for each dataset, I suggest that the authors write a summary showing advantages and disadvantages of the methods.
[R1] Nonlinear neural network mixture models for fractional abundance estimation in AVIRIS hyperspectral images, JPL, 2004
[R2] Support vector regression and synthetically mixed training data for quantifying urban land cover, RSE, 2013
[R3] A novel spectral unmixing method incorporating spectral variability within endmember classes, TGRS, 2016
Reviewer 3 Report
In this manuscript, a supervised method for nonlinear hyperspectral unmixing is proposed, whose contribution is to learn the mapping from the actual spectra to the corresponding linear spectra, composed of the same fractional abundances. I have the following comments: 1) As we know, neural network, Gaussian process, and kernel ridge regression are the traditional machine-learning methods. Although the mapping idea to solve the nonlinear unmixing is feasible, the solid contribution in theory is relatively limited. As such, the mechanism and rational for the mapping idea should be theoretically analyzed. 2) The proposed method assumes that a training set of spectral reflectances and ground-truth information about the endmembers and fractional abundances is available. However, the training set is very difficult to obtain. On the one hand, there is the possibility of no pure pixels in the observed hyperspectral data. On the other hand, the endmember variability will often happen. I am wondering how to address this issue for the proposed method in practice. 3) Please analyze the influence of the mapping accuracy to the final unmixing performance. 4) In Section 3, the convincing experiments should be provided for validating the effectiveness of the proposed method, for example, the robustness to the noise, the degree of nonlinear mixing, and different numbers of endmembers, and so on. 5) Please proofread the whole manuscript for revising the typos and syntax errors, as well as the reference format.